# History of the Wastewater Assessment of Polio and Non-Polio Enteroviruses in the Slovak Republic in 1963–2019

**DOI:** 10.3390/v14081599

**Published:** 2022-07-22

**Authors:** Renata Kissova, Katarina Pastuchova, Viera Lengyelova, Marek Svitok, Jan Mikas, Cyril Klement, Shubhada Bopegamage

**Affiliations:** 1Department of Medical Microbiology, Regional Public Health Authority Banska Bystrica, Cesta k Nemocnici 25, 97401 Banska Bystrica, Slovakia; renata.kissova@vzbb.sk (R.K.); klement@vzbb.sk (C.K.); 2National Reference Laboratory of Poliomyelitis Public Health Authority, Trnavska Cesta, 82102 Bratislava, Slovakia; katarina.pastuchova@uvzsr.sk; 3Department of Medical Microbiology, Regional Public Health Authority Senny Trh, 82009 Kosice, Slovakia; lengyelova@ruvzke.sk; 4Faculty of Ecology and Environmental Sciences, Technical University in Zvolen, 96001 Zvolen, Slovakia; svitok@tuzvo.sk; 5Public Health Authority, Trnavska Cesta, 82102 Bratislava, Slovakia; jan.mikas@uvzsr.sk; 6Faculty of Public Health, Slovak Medical University, Limbova 12, 83303 Bratislava, Slovakia; 7Faculty of Medicine, Enterovirus Laboratory, Institute of Microbiology, Slovak Medical University, Limbova 12, 83303 Bratislava, Slovakia

**Keywords:** enterovirus, polio surveillance in Slovak Republic, wastewater assessment, non-polio enteroviruses

## Abstract

We describe the genesis of poliovirus (PV) and non-polio enterovirus (NPEV) surveillance program of sewage wastewaters from its inception to the present in the Slovak Republic (SR). Sampling procedures and evolution of the methodology used in the SR for the detection of PVs and NPEVs are presented chronologically. For statistical data processing, we divided our dataset into two periods, the first period from 1963 to 1998 (35 years), and the second period from 1999 to 2019 (21 years). Generalized additive models were used to assess temporal trends in the probability of occurrence of major EV serotypes during both periods. Canonical correspondence analysis on relative abundance data was used to test temporal changes in the composition of virus assemblages over the second period. The probability of occurrence of major viruses PV, coxsackieviruses (CVA, CVB), and Echoviruses (E)) significantly changed over time. We found that 1015 isolated PVs were of vaccine origin, called “Sabin-like” (isolates PV1, PV2, PV3). The composition of EV assemblages changed significantly during the second period. We conclude that during the whole period, CVB5, CVB4, and E3 were prominent NPEVS in the SR.

## 1. Introduction

Enteroviruses (EV) are non-enveloped viruses with a single-stranded RNA of positive polarity. In the last century, during the 1950s, 19 antigenic types of Coxsackie A viruses (CVA), 5 antigenic types of Coxsackie B viruses (CVB), 19 antigenic types of “enteric cytopathic human orphan viruses” (ECHO viruses recently designated as E), and 3 types of polioviruses (PV) were known in this group [1]. During and after the 1960s and 1980s, the classification of EVs was revisited: E10 was reassigned to reoviruses, E8 was annulled, E28 was reassigned to rhinoviruses, CVA23 was renamed as E9, and E22 and EV23 were designated to parechoviruses [2]. In addition, 11 new EV serotypes were identified, from EV68 to EV78. Furthermore, EV72 was reclassified as a hepatovirus and the causative agent of viral hepatitis A [2,3,4]. According to the current taxonomy, EVs belong to the family *Picornaviridae*, genus *Enterovirus (EV)*, which includes 15 species, 7 of which are human pathogens. Currently, human EVs are classified into seven species: *EV-A* (25 serotypes), *EV-B* (63 serotypes), *EV-C* (23 serotypes), *EV-D* (5 serotypes), *Rhinovirus A* (80), *Rhinovirus B* (32), and *Rhinovirus C* (57) [4]. PVs (three serotypes) belong to the *EV-C* species.

Poliomyelitis is currently rare even in endemic countries because of the Global Poliomyelitis Eradication Program. In Europe, PVs have been eradicated since 2002. PV2 (all types in the year 2015) and PV3 (only the wild-type in the year 2019) were officially certified as globally eradicated viruses [5]. The last evidence of PV1 transmission was recorded in 2016; Afghanistan and Pakistan are polio-endemic countries where the transmission of indigenous wild poliovirus type1 (WPV1) has not been interrupted [6]. In Afghanistan, wild polioviruses type1 (WPV1) cases with an onset of paralysis were reported in 2021, (Kunduz and Ghazni provinces) and 2022. In Pakistan, WPV1 cases were reported in the year 2022 [7]. 

### 1.1. Background of the Slovak Polio Surveillance Program

EVs are excreted mainly in the feces for several weeks after the acute phase of the infection; consequently, they enter the environment, mainly wastewater, but also natural lakes used for bathing. They are resistant to low pH. They can persist in wastewater for some time and can be detected [8]. Surveillance of wastewater for the presence of EVs in the Slovak Republic (SR) (then still part of Czechoslovak Socialist Republic/Czechoslovakia (CSR)) began in the 1960s. The testing of wastewater first began in Bratislava (1963), West Slovakia, followed by Kosice (1965), East Slovakia, and Banska Bystrica (1968), Central Slovakia [9]. Mandatory vaccination of children against poliomyelitis, with the Salk’s inactivated polio vaccine (IPV), came into effect in 1957 in the former CSR [10], and since 1960, the live attenuated Sabin vaccine (OPV) was used for a definite period [11]. Since 1970, throughout the SR (then part of the CSR) wastewater examinations were carried out at various locations seasonally, at irregular intervals. The samples were collected from different facilities, most often schools, kindergartens, and boarding schools. Sampling points were selected mainly in the housing areas of larger cities. The number of sampling points changed over time and became stabilized around 1999 when the monitoring of wastewater for the presence of EVs became widespread and systematic. 

In 1988, the World Health Organization (WHO) announced a poliomyelitis eradication program [12]. In accordance with the WHO memorandum, a National Reference Center (NRC) for poliomyelitis was established by the Ministry of Health of SR in 1990 [13]. In 1998, this laboratory received accreditation as a “WHO Euro Polio Laboratory”. Following the NRC for poliomyelitis in Bratislava, two other virological laboratories in the SR, Banska Bystrica and Kosice, were included. The activities of these laboratories cover the entire territory (West, Central, and East) of the SR. 

Since 1988, environmental surveillance of PVs has been included as complementary surveillance for the monitoring of acute flaccid paralysis (AFP) in children under 15 years of age under the WHO Global Polio Eradication Initiative (GPEI) program [8,14,15]. The last WPV was isolated in the SR in 1960 (WPV1 in 1960, WPV2 in 1959, WPV3 in 1959, in wastewater and clinical samples) [9]. After the isolation of vaccine-derived PV (VDPV type 2) in the years 2003–2005 in the West region of SR (area Bratislava-Vrakuna, Skalica) [16,17], changes in the SR vaccination program were implemented. In the year 2005, the Sabin’s live oral polio vaccine (OPV) was replaced by the inactivated (IPV) Salk’s vaccine. All European countries gradually replaced OPV with IPV as a consequence of the eradication plan in the Euro region (June 2002). 

### 1.2. Present Status

A total of 47 sampling points exist currently in the SR. These are mainly at wastewater treatment plants in selected district towns, regional towns, and refugee camps. The network of sampling points covers the entire territory of the SR, and the monitored area includes about 60% of the population [18].

This study is an overview of the prevalence of PVs and other EVs in wastewater in the SR from 1963 to 2019 (56 years) and maps the changes and trends in the distribution of EVs in the SR.

## 2. Materials and Methods

### 2.1. Development of the Methodology

Over the years, methods of sampling and investigation of wastewater have changed (Appendix A). According to the available records from annual reports [9] since the year 1963, 1–5 L of wastewater was collected and filtered through cotton gauze; the concentration method included extraction with chloroform [19,20,21,22]. In 1968, due to the low yield of virus, the concentration method was changed, and aluminum hydroxide was added [23]. In 1970, the concentration method was changed again, back to the original one. Later, the flocculation method of processing was used, but the absence of a standard methodology was emphasized [9,24].

In 1986, the processing of water samples in large volumes (10 l) by the flocculation method with the use of aluminum sulphate was introduced nationwide [25]. This method was used to examine waters that were virus-positive as part of the indicative verification for the presence of virus indicators in waters, coliphage detection [25]. Later, the number of wastewater examinations was reduced, and the indicative water examination for the presence of coliphages was abandoned [9].

In 1994, the NRC for poliomyelitis began testing treated wastewaters using the two-phase separation method with dextran and polyethylene glycol (PEG), which was published in the WHO manual in 1990 [26]. Based on the regulation of the Chief Hygienist of the SR (end of 1998), the first schedule for wastewater collection developed in the West region of the SR was later used throughout the SR. From 1999 to the present day, the methodology of wastewater collection and treatment has been uniform in the SR, always in accordance with WHO recommendations [15]. Appendix A shows the overall changes in the methodology.

### 2.2. Present Methodology

Wastewater monitoring in the SR for the presence of PVs and other EVs is carried out according to the regulation of the Chief Hygienist of the SR—“Monitoring the circulation of PVs and other EVs in the external environment”. This regulation is updated annually, and the sampling is carried out by regional public health authorities in selected districts and county towns. Water is collected from stably designated sampling points: 42 sampling points from wastewater treatment plants in 39 districts and 3 regional cities and 5 refugee camps. Samples (a volume of 1 l) are taken just before the wastewater is discharged into the treatment plant (from 3–5 places, from a depth of about 30 cm below the surface) at each sampling point at approximately 2-month intervals, i.e., 6 times a year. 

After processing the wastewater samples, virus isolation on cell culture is carried out [26]. Since 1962, monkey kidney cells and human diploid embryonic lung cells (LEP) (from the Institute of Serum and Vaccines, Prague, CZ) have been used [9,27,28,29]. Other cell cultures have been used, such as HeLa cells, chicken fibroblasts, buffalo green monkey kidney (BGM) cells, and human amniotic cells [9], all obtained from the Institute of Serum and Vaccines, Prague, CZ. Lately, in the WHO manual from 1990 and the 1997 update [26], the human rhabdomyosarcoma-derived (RD) and human epidermoid carcinoma (Hep2) cell lines have been included, and these were recommended for the isolation of PVs and NPEVs. The current WHO manual [27] recommends two types of cell lines to investigate samples potentially containing PVs, i.e., genetically modified mouse cell lines with the human poliovirus receptor (L20B) and RD cell lines. The 2015 WHO EV surveillance guidelines [14] recommend the use of Hep2, RD, and African green monkey kidney cells (Vero cells) cell lines for the diagnosis of EVs. Currently, the cell lines L20B, RD, Hep-2 and Vero are used for PV and NPEV isolation in the SR. 

Presently, wastewater is treated with a two-phase separation method using dextran and PEG. Two eluates (two different phases) are obtained from each water sample. Then, 24–48 h monolayers of RD and L20B cell cultures (some laboratories additionally use Hep2 cell lines) are separately infected with 0.5 mL of each phase/bottle. A minimum of two 50 mL MD bottles (previously glass, now single-use Greiner Cellstar, Maybachstraße, Germany) of L20B and one 50 mL bottle of RD are used, as described in the manual [15]. Microscopic monitoring of cell cultures is performed daily, for 5–7 days. A minimum of two blind passages are performed. If the cytopathic effect is observed on RD cells, cross-passaging of RD isolates is carried out by further passaging in L20B cells. The identification of PVs and other NPEVs is conducted using the virus neutralization test (VNT) with WHO approved type-specific antisera, the Lim and Benyesh–Melnick pool, and PCR methods, following WHO recommendations [14,15,17,18,27]. 

### 2.3. Data Analysis

Until 1999, the sampling points were not assigned to particular locations, and the times of collection differed [9]. For this reason, in the statistical processing, we divided our dataset into two groups, the first from 1963 to 1998 (35 years), and the second from 1999 to 2019 (21 years). 

Generalized additive models (GAMs) [30] with binomial distribution and logit link function were used to assess temporal trends in the probability of occurrence of major groups during both periods. In the second period, finer taxonomic resolution allowed us to explore temporal patterns at the strain level. In the GAMs, for a given virus, the probability of occurrence was fitted with thin plate regression spline smoother of time to allow for potential non-linear trends. We carefully checked the model residuals for temporal autocorrelation. In several GAMs, significant serial dependence was detected. These models were subsequently rebuilt, including the autoregressive–moving average (ARMA) correlation structure. A sequence of ARMA GAMs was fitted to each response, and the model of the lowest ARMA order with a non-significant autocorrelation function was chosen as the most parsimonious one. The significance of the GAMs was assessed using Wald tests [31].

Since more detailed information on virus composition was available for the period 1999–2019, we used multivariate analysis to provide an overview of temporal changes in virus composition during this period. Canonical correspondence analysis (CCA) [32] was used to test for temporal changes in the composition of EV assemblages over the last two decades. The statistical significance of temporal trends was tested using a randomization test. Since the data were sampled in a temporal sequence, we restricted the randomization scheme to cyclic shifts along time [33].

The analyses were performed in R (R development Core Team 2019) [34] using the libraries ggplot2 [35], mgcv [30], and vegan [36]. 

## 3. Results

During the first period years 1963 to 2019, a total of 19,238 wastewater samples were examined. We isolated and identified 4881 (25 to 37%) EVs. Of these, 1062 were PVs, WPVs were not found. We found that 1015 isolated PVs were of vaccine origin, called “Sabin-like” (isolates PV1-SL 285, PV2-SL 293, PV3-SL 257). In the period 2003–2005, 47 vaccine-derived PV2 (cVDPV2) strains were isolated in the territory of the capital Bratislava (Vrakuna region) and in Skalica, both in the West region of the SR as seen in the Appendix A Even after increasing the environmental and clinical surveillance and an extensive epidemiological inquiry, the persons excreting the virus could not be identified [16]. Since 2005, vaccine strains of PVs have only been detected sporadically in wastewater. The last detection occurred in 2015 (PV1-SL in 2007, PV2-SL in 2013, PV3-SL in 2015).

Following a change in vaccination strategy, vaccine strains of polioviruses disappeared in the wastewater and were replaced by other EVs, most notably CVB, especially CVB5 [17,18]. Appendix A show the available information about the yearly identified strains and locations of the origins of the samples from the West (Appendix A), Central (Appendix A) and East (Appendix A) regions.

The GAMs showed a sharp decline in the occurrence of CVB (estimated degrees of freedom-edf = 3.7, F = 84.2, *p* < 0.0001) and E (edf = 3.7, χ^2^ = 92.5, *p* < 0.0001) from 1963 to 1970 (Figure 1a) and their subsequent increase during the second period (CVB: edf = 2.9, χ^2^ = 69.8, *p* < 0.0001; E: edf = 3.9, χ^2^ = 39.0, *p* < 0.0001) (Figure 1b). In contrast, PV showed three peaks of higher occurrence around 1970, 1988, and 2004 (1st period: edf = 4.0, F = 211, *p* < 0.0001; 2nd period: edf = 4.0, F = 995, *p* < 0.0001). The occurrence of the CVA group was generally low and showed a weakly significant declining trend during the first period (edf = 2.1, F = 4.5, *p* = 0.0144) and a non-significant pattern in the second period (edf = 2.7, χ^2^ = 6.4, *p* = 0.105). The trends detected at the group level during the second period were supported by the strain-level analysis (Figure 2). The most pronounced trend was the recent significant increase in CVB5 occurrence probability (edf = 3.6, F = 556.5, *p* < 0.0001). Other commonly appearing strains were CVB4 (edf = 3.9, χ^2^ = 20.4, *p* = 0.0004) and E3 (edf = 3.5, F = 13.0, *p* < 0.0001).

The CCA showed that the composition of EVs assemblages changed significantly over the last two decades (pseudo-F = 3.9, *p* = 0.0238). The first ordination axis accounted for 12.6% of the variability in the composition data and represents the long-term shift from assemblages dominated by PVs in the years after 2000 to the dominance of CVs in the years after 2010 (Figure 3). While PV1, PV2, PV3, CVA21, and E20, that were clustered in the left part of the ordination space, were typical of the early 2000s, CVB5, CVB4, E25m and CVA16 gained more dominance later. The second axis accounted for 19.2% of the variability and represents short-term shifts in the composition of virus assemblages. The assemblage composition was relatively constant during the last decade, as apparent from highly clustered sample scores in the right part of the ordination space. In contrast, considerable heterogeneity and sudden year-to-year changes occurred during the early 2000s, as evidenced by the large dispersion of the samples in the left part of the ordination space.

## 4. Discussion

For more than 50 years, PVs and other NPEVs have been monitored in wastewater in the SR. The monitoring is also associated with the surveillance of paralytic diseases caused by EVs and the monitoring of the effectiveness of vaccination against poliomyelitis, which began in the SR in 1960. In the late 1980s, this surveillance became part of the global poliomyelitis surveillance program. After 2000, as the vaccination strategy gradually changed worldwide and oral vaccination with live attenuated OPV was changed to intramuscular vaccination with an inactivated IPV vaccine, the prevalence of vaccine and vaccine-derived PVs in sewage declined, and we observed an increase, in NPEV isolates. 

Our study has limitations. Historical data on the EV strains isolated as well as the methods used were obtained from archival data of laboratories and their annual reports. These data may not have been complete in all laboratories. Different types of cell cultures were used during the first study period (1963–1998), and different methodologies were employed: both factors may have influenced the number and type of viruses isolated, especially NPEVs. In the second period (1999–2019), the same types of cell cultures and identical methodologies were used during the entire study period and in all three laboratories. This also was the main reason why the two time periods were also separated. 

The other limitation is the inability to make sufficient comparisons with other countries. We did not find any observational study about long-term evaluations of EVs in wastewater in the literature. At the same time, the viral strains evaluated by us were obtained only by classical cell culture methods, whereas most of the compared studies also reported results of EVs detection by molecular biology methods, so that strains that are more difficult to propagate in cell culture, e.g., CVA, EV-D68 or EV-A71, are not well represented. 

The prevalence and spectrum of occurrence of individual NPEVs have varied both over the years and in different European countries, as shown in a recent 3-year study of EVs prevalence in clinical samples in 24 European countries [37].

While vaccine strains of PVs (vaccine and vaccine-derived strains) dominated the wastewater in the SR from the early 1970s until the turn of the millennium, the situation has been changing dramatically in favor of NPEVs since the early 2000s. The emergence of PVs is observed to be recurring, with peaks around 1970, 1988, and 2004. The peak of PVs around 2004 was due to the emergence of PV2-VDPVs in sewage around Bratislava [16] and to the increased number of sample collections to check the VDPV isolates during this period (Figure 1b and Figure 2).

After 2005, following a change in the vaccination strategy, there was a sharp decline in the incidence of PVs, which started to appear only sporadically in wastewater. The last detection was in 2015 (PV1-SL in 2007, PV2-SL in 2013, PV3-SL in 2015). 

Es also appeared periodically, with their peaks slightly replicating those of PVs, between 1980 and 1990 and around 2005. The next peak appeared around 2014, indicating an upsurge approximately every 10–15 years (Figure 1). The cyclical occurrence of Es has also been described by other studies [37]. However, the occurrence of these viruses in wastewater in Slovakia has a long-term increasing trend. The most frequently occurring E in Slovakia is E3.

The emergence of CVBs after 2000 was significant (Figure 1). Of these, CVB5 is the most common and currently dominant in Slovak wastewater (Figure 2). CVB5 was also highly dominant during a 1-year monitoring of wastewater in France (Clermont-Ferrand area) [38] and during a 13-year monitoring in Italy [39]. 

A 3-year study of EVs prevalence in clinical samples in the European region showed that CVA6 was the most prevalent strain (13.46%) [37]. Our study shows that CVB5 was present in 8.18% of samples in Slovakia [37], whereas CVA6 was not isolated from either sewage or clinical samples during this period. The prevalence of CVAs in the SR was previously low in the sewage, although these viruses have also increased over the last 20 years (Figure 1).

The highest number of isolates was always retrieved in late summer and autumn, which is a typical period for EVs [40]. Parallelly, a decrease in the heterogeneity of the isolated strains can be observed over the study period, mirrored by highly clustered sample scores in the right part of the CCA ordination (Figure 3).

Available data from other countries regarding EV strains in wastewater show that in Italy (in 2006–2010 and 2005–2018), CVB5, E6, and CVB4 were the most frequently occurring EVs in wastewater [39,41]. 

In observations from the Republic of Moldova (years 2002–2019), vaccine strains of PVs were isolated in 53% of positive wastewater samples. Among the NPEVs, E30, E11, E6, and CVB1-6 were dominant [42]. In the Russian Federation, in the vicinity of Moscow, vaccine strains of PVs were isolated in 43% of virus-positive wastewater samples in 2004–2017. The most detected NPEVs were E7, E11, E6, CVB5, E3, and E19 (39). In a 1-year retrospective study from the USA (2010–2011), CVA1, 6, 19, CVB3 and 5, E9, and EV97 were the predominant viruses [43]. 

Different serotypes of the species EV-B are also the most frequently found viruses in wastewater in other countries all over the world [38,39,41,42,44,45]. 

## 5. Conclusions

After a radical reduction in the prevalence of PVs in the population (by eradication of both WPV strains and vaccine PVs, a consequence of the switching from a live to an inactivated polio vaccine), an increase in the NPEVs was observed. At the same time, we found a decrease of heterogeneity in the composition of viral strains in wastewaters in the SR. Besides the vaccine PV strains, the most frequently occurring viruses during the whole study period were CVB5, CVB4, and E3. With the recent global challenges such as the COVID-19 pandemic, the influx of refugees, and the climate and environmental changes, we believe that testing for EVs should proceed and that existing networks of laboratories formed over the years should take a vital part in this process. 

## Figures and Tables

**Figure 1 viruses-14-01599-f001:**
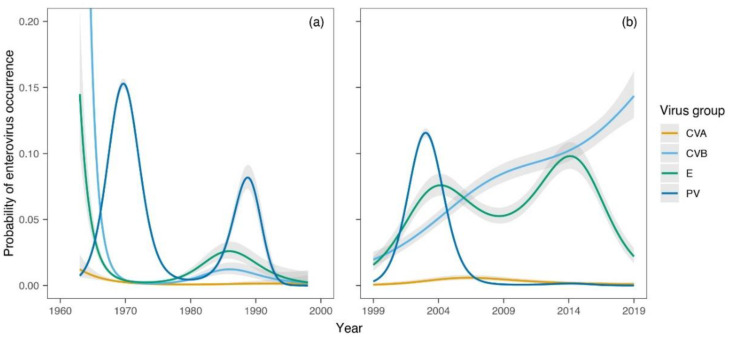
(**a**) Generalized additive models (GAM) showing temporal trends in the probability of enteroviruses (EV) strains, coxsackievirus A (CVA), CVB, echoviruses (E), and polioviruses (PV) occurrence during the first period (1963–1998). (**b**) Temporal trends in the probability of EV groups occurrence during the second period (1999–2019). GAM-based estimates (lines) are displayed along with their 95% confidence intervals.

**Figure 2 viruses-14-01599-f002:**
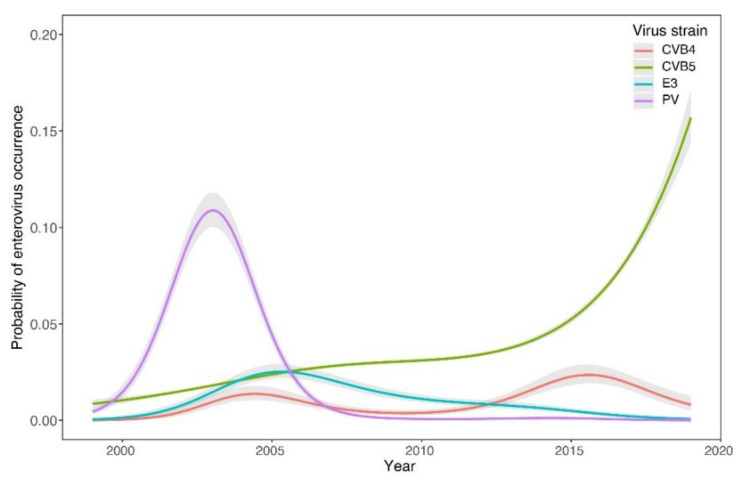
Temporal trends in the probability of enterovirus (EV) strains, coxsackievirus A (CVA), CVB, echoviruses (E) and polioviruses (PV) occurrence during the second period (1999–2019). Generalized additive models (GAM)-based estimates (lines) are displayed along with their 95% confidence intervals.

**Figure 3 viruses-14-01599-f003:**
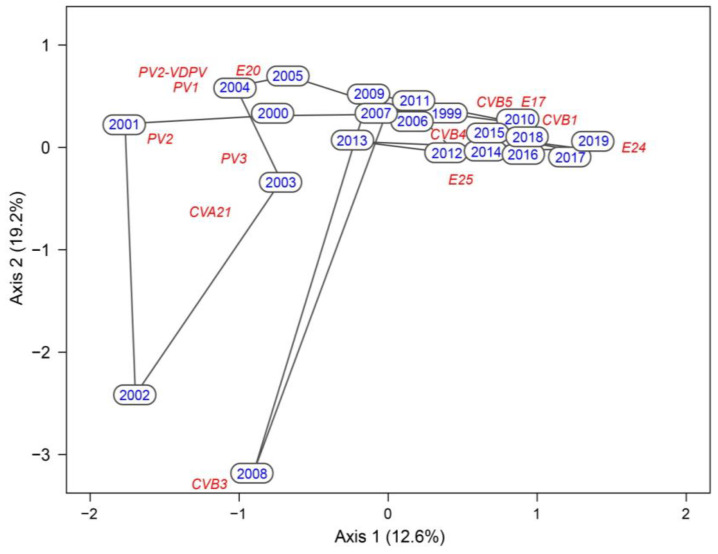
Canonical correspondence analysis (CCA) ordination plot showing temporal changes in the assemblage composition of enterovirus (EV) strains, coxsackievirus A (CVA), CVB, echoviruses (E), polioviruses (PV), and vaccine-derived poliovirus (VDPV) during the second period (1999–2019). Only the most frequent strains with the best fit to the long-term temporal trend (first axis) are displayed (in italics). Sample scores are labelled by years (in blue), and the strain scores, representing the maxima of relative abundances, are shown in red. The ordination plot is scaled symmetrically; variation explained by the ordination axes is displayed in parentheses.

## Data Availability

At the Authority of the Public Health and Regional Authorities of Public Health of the Slovak Republic.

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
