# Peer review of "History of the Wastewater Assessment of Polio and Non-Polio Enteroviruses in the Slovak Republic in 1963–2019"

_viruses, 2022, doi:10.3390/v14081599_

Round 1

Reviewer 1 Report

The manuscript by Kissova R. et al. provides a good and comprehensive historical overview covering the detection of enteroviruses in wastewater over a period of more than 50 years. Wastewater testing is an important additional tool for monitoring a country's polio-free status. Especially in IPV using countries, it can also be used as an early warning system. Silent circulation of polioviruses can occur in under-vaccinated communities.

Currently 47 wastewater treatment plants (district level, regional cities, refugee camps), covering 60% of population, are sampled. Sampling is carried out through regional public health authorities on a bi-monthly interval.

Some minor comments/questions:

Abstract

-          “The last WPV was isolated in the SR in 1960” - should be deleted from the abstract, as these data are not part of the evaluated data and were previously collected.

Introduction

-          Enteroviruses infecting humans are assigned to four species, Enterovirus A – Enterovirus D; enteroviruses („namely EV68 to EV78“) are new types (not species),

-          ”serotypes“ -> types should be used instead, as not all viruses can be neutralized be specific antisera

-          Eradication of PV2 covers all types, whereas only wild type for PV3

-          Last case in Nigeria was in 2016

-          Detailed numbers from PAK and AFG can be deleted (as there are frequent changes)

-          „bathing water”- do you mean bathing/swimming lake/pond?

-          “monitoring of wastewater for the presence of enteral viruses”- should be more precise: which viruses were included (adeno-, rota- noroviruses?), or do you mean NPEV?

Methods

-          Development of the methodology-> focus more on current method (perhaps the old methods could be presented in a table for better clarity?): it could be made clearer which methods have been used in the past and which are current (current algorithm could be explained in more detail - number of bottles inoculated, duration of observation).

-           “This method was used to examine waters that were positive in the verification process for the verification known as the presence of virus indicators in the waters - detection of coliphages”- should be explained easier

-          Ref [27] may be not the correct one here, as this is for investigation of stool samples, wastewater collection etc. is not described here. Specific guidelines for environmental samples exist.

Results

-          the data analysis should be presented in a more compact and understandable way

-           “PV were replaced by other enteral viruses”- does it mean, that NPEV were not found before?

-          Can the changes in the occurrence of CVB (decline during the first and increase during the second period) be explained by the use of other/different cell lines?

-          The low detection of the CVA is probably not surprising, as not all of them are cultivable.

-          Although CCA is a recognised statistical method, the data in figure 3 are not self-explanatory. Therefore, the figure should be better explained for non-statisticians. What do the types in red mean?

 Discussion

-          “Paralytic disease caused by viruses “– which viruses are tested (EV?)

-          “the prevalence of vaccine and vaccine-derived polioviruses in sewage declined”- From a global point of view, this should be formulated more cautiously, since (especially in Africa) a lot of VDPVs are currently detected from environmental samples

-          Is it legitimate to compare NPEV detections in clinical samples with detections in wastewater, are there correlations? Did you compare your data from wastewater and clinical samples?

-          decrease in the heterogeneity of isolated strains- I do not see it from fig.3. Maybe the presentation of these results as stack chart (viruses per year) is more understandable?

 The limitations were well elaborated.

 Supplementary material (figure) should be omitted.

Reviewer 2 Report

The authors report a retrospective assessment study of enterovirus isolates, including polioviruses, detected in sewage samples in the Slovak Republic since 1963. The period of this study is exceptional. The data reported by the authors are important to the community. This study is timely as poliovirus-derived vaccine strains were recently detected in London.

I have appended a number of comments to improve the manuscript.

Abstract

The abstract section should revised and structured to reflect the data and methods reported/used in the manuscript. In particular, please indicate clearly which data were submitted to statistical analyses and which statistical analyses were done.

The sentence indicating the isolation of WPV should be deleted; these viruses were reported outside the study period.

1,062 PVs were reported and 1,015 were identified as Sabin-like: what are the origins of the other 47 PVs?

“chronically”, please explain.

Section 1 Introduction

Literature review as written in this section is concise and relevant.

Please explain the classification of poliovirus isolates: Sabin-like viruses and cVDPVs.

It is difficult to figure out the data used in the study. Please indicate clearly the aim of the study and which data were collected (see below).

Line 58, typo “type1”  

Line 77, “enteral”: do you mean enteroviral or enteric?

Line 79, “eradicate”: eradication?

Section 2

The “crude data” analysed in the present study (i.e. the data recorded from the reports reviewed from the 1960s) should be summarized and presented in the manuscript. For instance, the authors should produce a table indicating for the different periods, the number of samples tested, the number of sampling points, WW sample volume, the number of PV and NPEV isolated, the cell lines used over time, virus concentration method, virus identification, etc…

The overall methodology used in the study seems sound. However, the statistical analyses applied to the data need to be reviewed by an expert in the field.

Line 150, are RD(A) different from RD? Please, explain.

Section 3

Provided the statistical analyses are valid, the results are comprehensively presented with clear figures. My main concern is related to the multiple changes that were introduced in the methodology since 1963. How the influence of these changes on the results were managed in the statistical analyses. For instance, is it possible to assess the influence of using different cell lines on the EV types detected over the two periods considered (or over time)?

Lines 184 – 186. The sentence should be deleted and included in the introduction, because detection of wild type PVs was performed before the study period.

Line 195. Please check “enteral”.

Figure 1. Check the legend to the figure and define the virus groups.

Figure 2. Please, check the typo in the title of y-axis.

Figure 3. Define the abbreviations (PV2-VDPV, E20…) in the legend to the figure.

Section 4

Discussion is reasonable and sound, however I have a number of concerns.

Line 223. “were replaced”, do you really think that the data of virus occurrence in isolation tests should be interpreted in terms of replacement? Given the limitations of the study and the epidemiological patterns of EV infections, it is uncertain for instance to interpret the data as meaning that polioviruses were replaced by NPEVs, which implies (wrongly in my opinion) a causal link.

The discussion on the limitations of the study should be expanded, in particular regarding the changes in the cell lines used over the period and their patterns of sensitivity to the various EV types. Perhaps, the main limitation is a lack of comparison with the detection of NPEVs associated with clinical infections. I understand that it cannot be done for the entire period but it would be interesting to have this comparison at least for a limited period. For instance, does the increase in CVB detection in WW samples after 2014 coincide with an increase of CVBs in clinical data?

Line 230. “We did find” -> We did not find?

Line 235. “EV-A7” -> EV-A71?

Lines 266 – 267. Not sure to understand what does the sentence means: must this be interpreted as an epidemiological trend or an artifact? Please, comment.
